# Continuous Indeterminate Probability Neural Network

## Abstract

Currently, there is no mathematical analytical form for a general posterior, however, Indeterminate Probability Theory Anonymous (2024) has now discovered a way to address this issue. This is a big discovery in the field of probability and it is also applicable in various other fields. This paper introduces a general model called **CIPNN** – **C**ontinuous **I**ndeterminate **P**robability **N**eural **N**etwork, which is an **analytical** probability neural network with continuous latent random variables. Our contributions are Four-fold. First, we apply the analytical form of the posterior for continuous latent random variables and propose a general classification model (CIPNN). Second, we propose a general auto-encoder called CIPAE – Continuous Indeterminate Probability Auto-Encoder, instead of using a neural network as the decoder component, we first employ a probabilistic equation. Third, we propose a new method to visualize the latent random variables, we use one of N dimensional latent variables as a decoder to reconstruct the input image, which can work even for classification tasks, in this way, we can see what each latent variable has learned. Fourth, IPNN has shown great classification capability, CIPNN has pushed this classification capability to infinity. Theoretical advantages are reflected in experimental results.

## 1 Introduction

Although recent breakthroughs demonstrate that neural networks are remarkably adept at natural language processing Vaswani et al. (2017); Devlin et al. (2019); Ouyang et al. (2022), image processing He et al. (2016), neural networks are still black-box for human Buhrmester et al. (2019), cognitive scientists and neuroscientist have argued that neural networks are limited in their ability to represent variables and data structures Graves et al. (2016); Bottou (2011). Probabilistic models are mathematical descriptions of various natural and artificial phenomena learned from data, they are useful for understanding such phenomena, for prediction of unknowns in the future, and for various forms of assisted or automated decision making Kingma & Welling (2019).

Deep Latent Variable Models (DLVMs) are probabilistic models and can refer to the use of neural networks to perform latent variable inference Kim et al. (2018). Currently, the posterior calculation is regarded as intractable Kingma & Welling (2014; 2019), and the variational inference method is used for efficient approximate posterior inference Kingma & Welling (2014); Titsias & Lázaro-Gredilla (2014); Rezende et al. (2014).

Indeterminate Probability Theory provides the analytical solution for any complex posterior calculation, and the first analytical probability model proposed based on it is called IPNN Anonymous (2024). However, IPNN need predefine the sample space of each discrete random variable (called 'split shape' in IPNN), it is sometimes hard to define a proper sample space for an unknown dataset. For CIPNN, the sample space of each continuous random variable is infinite, this issue will not exit in CIPNN.

The rest of this paper is organized as follows: In Sec. 2, related work of VAE and Indeterminate Probability Theory is introduced. In Sec. 3, we use a simple coin toss example to explain the core idea of CIPNN, and the proposed question cannot be analytically solved with current other probability theories. In Sec. 4, CIPNN is derived and the regularization method is discussed. In Sec. 5, CIPAE is derived and we propose a new method to visualize each latent variable. In Sec. 6, we discuss the training strategy, and two common training setups are discussed: CIPNN and CIPAE

are combined together for better evaluation of classification and auto-encoder tasks. In Sec. 7, CIPNN and CIPAE are evaluated and the latent variables are visualized with our new proposed method. Finally, we put forward some future research ideas and conclude the paper in Sec. 8.

## 2 RELATED WORK

### 2.1 VAE

Modern machine learning and statistical applications require large scale inference in complex models, the inference models are regarded as intractable and either Markov Chain Monte Carlo (MCMC) Robert & Casella (2004) or variational Bayesian inference Jordan et al. (1999) are used as approximate solutions Titsias & Lázaro-Gredilla (2014). VAE Kingma & Welling (2014) proposes an estimator of the variational lower bound for efficient approximate inference with continuous latent variables. DARN method is generative auto-encoder capable of learning hierarchies of distributed representations from data, and their method applies to binary latent variables Gregor et al. (2013). In concurrent work of VAE, two later independent papers proposed equivalent algorithms Titsias & Lázaro-Gredilla (2014); Rezende et al. (2014), which provides an additional perspective on VAE and the latter work applies also the same reparameterization method. Two methods proposed by VAE are also used to realize our analytical solution: the reparameterization trick for making the model differentiable and the KL divergence term for regularization.

VAEs have been used for many tasks such as image generation Razavi et al. (2019), anomaly detection Xu et al. (2018) and de-noising tasks Im et al. (2017) Boyar & Takeuchi (2023). The drawback of auto-encoder is its strong tendency to over-fit Steck (2020), as it is solely trained to encode and decode with as little loss as possible regardless of how the latent space is organized Yue et al. (2023b), VAE has been developed as an effective solutions Steck (2020); Bi et al. (2019), e.g. VAEs has been used in EEG classification tasks to learn robust features Yue et al. (2023a); Bethge et al. (2022); Bi et al. (2019); Bollens et al. (2022).

The framework of our CIPAE is almost the same as that of VAE, the only difference is that VAE uses neural network as the approximate solution of decoder, while CIPAE uses probabilistic equation as the analytical solution of decoder.

### 2.2 INDETERMINATE PROBABILITY THEORY

Indeterminate Probability Theory proposes a new perspective for understanding probability theory by introducing Observers and treating the outcome of each random experiment as indeterminate probability distribution, which leads to probability calculations being a combination of ground truth and observation errors. Here is a short summary of Indeterminate Probability Theory:

Special random variable $X \in \{x_1, x_2, \ldots, x_n\}$ is defined for random experiments, and $X = x_k$ is for $k^{th}$ experiment, so $P(x_k) \equiv 1$. Random variable $Y \in \{y_1, y_2, \ldots, y_m\}$ is a general discrete variable (continuous variable is also allowed), $P(y_l|x_k) = y_l(k) \in [0, 1]$ is the indeterminate probability to describe the observed outcome of sample $x_k$. $P^{\mathbf{z}}(y_l \mid x_t)$ is for the inference outcome of sample $x_t$, superscript $\mathbf{z}$ stands for the medium – N-dimensional latent random variables $\mathbf{z} = (z^1, z^2, \ldots, z^N)$, via which we can infer $Y = y_l, l = 1, 2, \ldots, m$.

The analytical solution of the posterior is as bellow Anonymous (2024):

$$P\left(y_l \mid z^1, z^2, \ldots, z^N\right) = \frac{\sum_{k=1}^{n} \left(P\left(y_l \mid x_k\right) \cdot \prod_{i=1}^{N} P\left(z^i \mid x_k\right)\right)}{\sum_{k=1}^{n} \prod_{i=1}^{N} P\left(z^i \mid x_k\right)} \tag{1}$$

And inference probability with the posterior is

$$P^{\mathbf{z}}\left(y_l \mid x_t\right) = \int_{\mathbf{z}} \left(P\left(y_l \mid z^1, z^2, \ldots, z^N\right) \cdot \prod_{i=1}^{N} P\left(z^i \mid x_t\right)\right) \tag{2}$$

## 3 BACKGROUND

Before introducing CIPNN, we will use a simple coin toss example to demonstrate how to use Eq. (1) and Eq. (2) for continuous random variables, see Table 1.

Table 1: Example of coin toss.

| Random Experiment ID $X$ | $x_1$ | $x_2$ | $x_3$ | $x_4$ | $x_5$ |
| | $x_6$ | $x_7$ | $x_8$ | $x_9$ | $x_{10}$ |
| Ground Truth | $hd$ | $hd$ | $hd$ | $hd$ | $hd$ |
| | $tl$ | $tl$ | $tl$ | $tl$ | $tl$ |
| Record of Observer$_1$ $Y$ | $hd$ | $hd$ | $hd$ | $hd$ | $hd$ |
| | $tl$ | $tl$ | $tl$ | $tl$ | $tl$ |
| Equivalent Record $Y$ | 1, 0 | 1, 0 | 1, 0 | 1, 0 | 1, 0 |
| | 0, 1 | 0, 1 | 0, 1 | 0, 1 | 0, 1 |
| Record of Observer$_2$ $A$ | 0.8, 0.2 | 0.7, 0.3 | 0.9, 0.1 | 0.6, 0.4 | 0.8, 0.2 |
| | 0.1, 0.9 | 0.2, 0.8 | 0.3, 0.7 | 0.1, 0.9 | 0.2, 0.8 |
| Record of Observer$_3$ $z$ | $\mathcal{N}(3,1)$ | $\mathcal{N}(3,1)$ | $\mathcal{N}(3,1)$ | $\mathcal{N}(3,1)$ | $\mathcal{N}(3,1)$ |
| | $\mathcal{N}(-3,1)$ | $\mathcal{N}(-3,1)$ | $\mathcal{N}(-3,1)$ | $\mathcal{N}(-3,1)$ | $\mathcal{N}(-3,1)$ |

Where $hd$ is for head, $tl$ is for tail. And condition on $x_k$ is the indeterminate probability, e.g. $P(Y = hd|X = x_3) = 1$, $P(A = tl|X = x_6) = 0.9$ and $P(z|X = x_8) = \mathcal{N}(z; -3, 1)$.

**Observer$_1$**   Let's say, Observer$_1$ is an adult and record the outcome of coin toss always correctly, so the probability of $Y$ can be easily calculated with the general probability form:

$$P(Y = hd) = \frac{\text{number of } (Y = hd) \text{ occurs}}{\text{number of random experiments}} = \frac{5}{10} \tag{3}$$

If we represent Observer$_1$'s record with equivalent form of $P(Y = hd|X = x_k)$, the probability is:

$$P(Y = hd) = \sum_{k=1}^{10} P(Y = hd|X = x_k) \cdot P(X = x_k) = \frac{5}{10} \tag{4}$$

**Observer$_2$**   Let's say, Observer$_2$ is a model, it takes the image of each coin toss outcome as inputs, and it's outputs are discrete probability distribution.

The Observer$_2$'s record probability is

$$P(A = hd) = \sum_{k=1}^{10} P(A = hd|X = x_k) \cdot P(X = x_k) = \frac{4.7}{10} \tag{5}$$

This calculation result is a combination of **ground truth** and **observation errors**.

**Observer$_3$**   Let's say, Observer$_3$ is a strange unknown observer, it always outputs a Gaussian distribution for each coin toss with a 'to-be-discovered' pattern. How can we find this pattern?

$$P(z) = \sum_{k=1}^{10} P(z|X = x_k) \cdot P(X = x_k) = \frac{5 \cdot \mathcal{N}(z; 3, 1) + 5 \cdot \mathcal{N}(z; -3, 1)}{10} \tag{6}$$

We get a complexer $P(z)$ distribution here, it's form is still analytical. And this distribution have two bumps, how can we know the representation of each bump mathematically? We need to use the Observer$_1$'s record $Y$. With Eq. (1) we have

$$P(Y = hd|z) = \frac{\sum_{k=1}^{10} P(Y = hd|X = x_k) \cdot P(z|X = x_k)}{\sum_{k=1}^{10} P(z|X = x_k)} = \frac{\mathcal{N}(z; 3, 1)}{\mathcal{N}(z; 3, 1) + \mathcal{N}(z; -3, 1)} \quad (7)$$

For next coin toss, let $P(z|X = x_{11}) = \mathcal{N}(z; 3, 1)$, With Eq. (2) and Monte Carlo method, we have

$$P^z(Y = hd|X = x_{11}) = \int_z \left( P(Y = hd|z) \cdot P(z|X = x_{11}) \right)$$

$$= \mathbb{E}_{z \sim P(z|X=x_{11})} \left[ P(Y = hd|z) \right] \approx \frac{1}{C} \sum_{c=1}^{C} P(Y = hd|z_c) \quad (8)$$

$$= \frac{1}{C} \sum_{c=1}^{C} \frac{\mathcal{N}(z_c; 3, 1)}{\mathcal{N}(z_c; 3, 1) + \mathcal{N}(z_c; -3, 1)} \approx 1, z_c \sim \mathcal{N}(z; 3, 1)$$

Where $C$ is for Monte Carlo number. In this way, we know that the bump with mean value 3 is for $Y = hd$. Note: this issue cannot be analytically solved with current other probability theories.

If we use a neural network to act as observer$_3$ to output multivariate Gaussian distributions, this is the core idea of our CIPNN and CIPAE model, and their forms are still analytical.

# 4 CIPNN

For neural network tasks, $X = x_k$ is for the $k^{th}$ input sample, $P(y_l|x_k) = y_l(k) \in [0, 1]$ is for the soft/hard label of train sample $x_k$, $P^{\mathbf{z}}(y_l \mid x_t)$ is for the predicted label of test sample $x_t$.

## 4.1 CONTINUOUS INDETERMINATE PROBABILITY

Figure 1 shows CIPNN model architecture, the neural network is used to output the parameter $\theta$ of certain distributions (e.g. Gaussian distributions) of multivariate latent continuous random variable $\mathbf{z} = (z^1, z^2, \ldots, z^N)$. And the latent space is fully connected with all labels $Y \in \{y_1, y_2, \ldots, y_m\}$ via conditional probability $P(y_l \mid z^1, z^2, \ldots, z^N)$.

For each continuous random variable $z^i$, the indeterminate probability (density function) is formulated as:

$$P(z^i \mid x_k) = p(z; \theta_k^i), i = 1, 2, \ldots, N. \quad (9)$$

Where $z$ is for a certain distribution with a generated parameter $\theta_k^i$.

Firstly, we substitute $P(y_l|x_k) = y_l(k)$ and Eq. (9) into Eq. (2) and Eq. (1), which gives us Eq. (10). Secondly, due to the complicated integration over $\mathbf{z}$, we derive an exceptional Eq. (11) because of $\int_{\mathbf{z}} \left( \prod_{i=1}^{N} p(z; \theta_t^i) \right) = 1$. Thirdly, before using Monte Carlo method Robert & Casella (2004) to make an approximate estimation, we use the reparameterization trick Kingma & Welling (2014): let $\varepsilon \sim p(\varepsilon)$ be some random noise, and define a mapping function $z = g(\varepsilon, \theta)$. Thus, $p(z; \theta_k^i)$ can be rewritten as $p(g(\varepsilon, \theta); \theta_k^i)$, see Eq. (12) and Eq. (13).

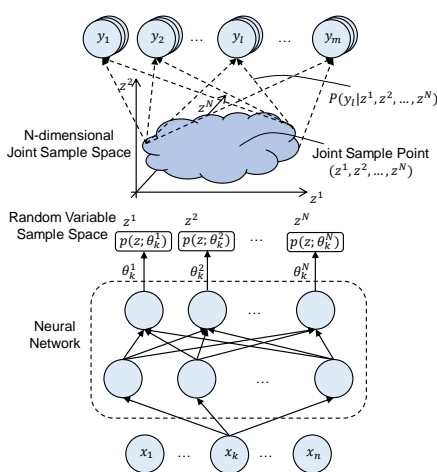

Figure 1: CIPNN – model architecture. Where $P(y_l \mid z^1, z^2, \ldots, z^N)$ is calculated with Eq. (1), not model weights.

$$P^{\mathbf{z}}\left(y_l \mid x_t\right) = \int_{\mathbf{z}} \left( \frac{\sum_{k=1}^n \left(y_l(k) \cdot \prod_{i=1}^N p\left(z; \theta_k^i\right)\right)}{\sum_{k=1}^n \left(\prod_{i=1}^N p\left(z; \theta_k^i\right)\right)} \cdot \prod_{i=1}^N p\left(z; \theta_t^i\right) \right) \tag{10}$$

$$= \mathbb{E}_{z \sim p\left(z; \theta_t^i\right)} \left[ \frac{\sum_{k=1}^n \left(y_l(k) \cdot \prod_{i=1}^N p\left(z; \theta_k^i\right)\right)}{\sum_{k=1}^n \left(\prod_{i=1}^N p\left(z; \theta_k^i\right)\right)} \right] \tag{11}$$

$$= \mathbb{E}_{\varepsilon \sim p(\varepsilon)} \left[ \frac{\sum_{k=1}^n \left(y_l(k) \cdot \prod_{i=1}^N p\left(g(\varepsilon, \theta_t^i); \theta_k^i\right)\right)}{\sum_{k=1}^n \left(\prod_{i=1}^N p\left(g(\varepsilon, \theta_t^i); \theta_k^i\right)\right)} \right] \tag{12}$$

$$\approx \frac{1}{C} \sum_{c=1}^C \left( \frac{\sum_{k=1}^n \left(y_l(k) \cdot \prod_{i=1}^N p\left(g(\varepsilon_c, \theta_t^i); \theta_k^i\right)\right)}{\sum_{k=1}^n \left(\prod_{i=1}^N p\left(g(\varepsilon_c, \theta_t^i); \theta_k^i\right)\right)} \right), \varepsilon_c \sim p\left(\varepsilon\right) \tag{13}$$

Take, for example, the Gaussian case: let $P\left(z^i \mid x_k\right) = \mathcal{N}\left(z; \mu_k^i, \sigma_k^{2,i}\right)$, $\theta_k^i := (\mu_k^i, \sigma_k^{2,i})$ and let $\varepsilon_c \sim \mathcal{N}\left(0, 1\right)$, we get the reparameterization function $g\left(\varepsilon_c, \theta_t^i\right) = \mu_t^i + \sigma_t^i \cdot \varepsilon_c$, Eq. (13) can be written as:

$$P^{\mathbf{z}}\left(y_l \mid x_t\right) \approx \frac{1}{C} \sum_{c=1}^C \left( \frac{\sum_{k=1}^n \left(y_l(k) \cdot \prod_{i=1}^N \mathcal{N}\left(\mu_t^i + \sigma_t^i \cdot \varepsilon_c; \mu_k^i, \sigma_k^{2,i}\right)\right)}{\sum_{k=1}^n \left(\prod_{i=1}^N \mathcal{N}\left(\mu_t^i + \sigma_t^i \cdot \varepsilon_c; \mu_k^i, \sigma_k^{2,i}\right)\right)} \right) \tag{14}$$

We use cross entropy as main loss function:

$$\mathcal{L}_{main} = -\sum_{l=1}^m \left(y_l(t) \cdot \log P^{\mathbf{z}}\left(y_l \mid x_t\right)\right) \tag{15}$$

## 4.2 REGULARIZATION

The sufficient and necessary condition of achieving global minimum is already proved in IPNN Anonymous (2024), which is also valid for continuous latent variables:

**Proposition 1** *For $P(y_l|x_k) = y_l(k) \in \{0, 1\}$ hard label case, CIPNN converges to global minimum only when $P\left(y_l|z^1, z^2, \ldots, z^N\right) \to 1$, for $\prod_{i=1}^N p\left(z; \theta_t^i\right) > 0$.*

*In other word, each N-dimensional joint sample area (collection of adjacent joint sample points) corresponds to an unique category. However, a category can correspond to one or more joint sample areas.*

According to above proposition, the reduction of training loss will minimize the overlap between distribution $\prod_{i=1}^N p\left(z; \theta_t^i\right)$ of each category. For Gaussian distribution, the variance will be close to zero, and the distribution of each category will be far away from each other. This will cause over-fitting problem Yue et al. (2023b); Steck (2020).

VAE uses a regularization loss to avoid the over-fitting problem Kingma & Welling (2014; 2019), and there are follow up works which has proposed to strengthen this regularization term, such as $\beta$-VAE Higgins et al. (2017); Burgess et al. (2018), $\beta$-TCVAE Chen et al. (2018), etc. In order to fix the over-fitting problem of CIPNN, we have a modification of VAE regularization loss:

$$\mathcal{L}_{reg} = \sum_{i=1}^N \left( D_{KL}\left(\mathcal{N}\left(z; \mu_t^i, \sigma_t^{2,i}\right) || \mathcal{N}\left(z; \gamma \cdot \mu_t^i, 1\right)\right)\right)$$

$$= \frac{1}{2} \sum_{i=1}^N \left(((1-\gamma) \cdot \mu_t^i)^2 + \sigma_t^{2,i} - \log(\sigma_t^{2,i}) - 1\right) \tag{16}$$

Where $N$ is the dimensionality of $\mathbf{z}$, regularization factor $\gamma \in [0, 1]$ is a hyperparameter and is used to constrain the conditional joint distribution of each category to be closely connected with each other, impact analysis of regularization factor $\gamma$ see Figure 8.

According to lagrange multipliers, we have the overall loss as

$$\mathcal{L} = \mathcal{L}_{main} + \beta \cdot \mathcal{L}_{reg} \tag{17}$$

## 5 CIPAE

For image auto-encoder task, we firstly transform the pixel value to $[0, 1]$ (Bernoulli distribution), and let $Y^j \in \{y_1^j, y_2^j\}_{j=1}^J$, where $J$ is the number of all pixels of one image. $P(y_1^j|x_k) = p_1^j(k) \in [0, 1]$, which describes the pixel value of image $x_k$ at $j^{th}$ position, and $P(y_2^j|x_k) = p_2^j(k) = 1 - p_1^j(k)$.

Substitute $P(y_l^j|x_k)$ into Eq. (14), we will get $P^{\mathbf{z}}\left(y_l^j \mid x_t\right), l = 1, 2$. In this way, the reconstructed image is formulated as:

$$\text{reconstructed image} := \left\{P^{\mathbf{z}}\left(y_1^j \mid x_t\right)\right\}_{j=1}^J \tag{18}$$

In addition, with one (or part) of N dimensional latent variables we can also reconstruct the input image, the reconstructed image is:[1]

$$\text{reconstructed feature} := \left\{P^{z^i}\left(y_1^j \mid x_t\right)\right\}_{j=1}^J \tag{19}$$

Where $i = 1, 2, \ldots, N$. In this way, we can see what each latent variable has learned.

Substitute Eq. (18) into Eq. (15), we can get a binary cross entropy loss:

$$\mathcal{L}_{main} = -\frac{1}{J} \sum_{j=1}^J \sum_{l=1}^2 \left(p_l^j(t) \cdot P^{\mathbf{z}}\left(y_l^j \mid x_t\right)\right) \tag{20}$$

And substitute the above loss into Eq. (17), we get the overall loss for auto-encoder training.

## 6 TRAINING

In this section, we will focus on the training strategy of Gaussian distribution.

### 6.1 TRAINING STRATEGY

Given an input sample $x_t$ from a mini batch, with a minor modification of Eq. (14):

$$P^{\mathbf{z}}\left(y_l \mid x_t\right) \approx \frac{1}{C} \sum_{c=1}^C \left(\frac{\max(H(\varepsilon_c), \epsilon)}{\max(G(\varepsilon_c), \epsilon)}\right) \tag{21}$$

Where stable number $\epsilon$ on the denominator is to avoid dividing zero, $\epsilon$ on the numerator is to have an initial value of 1. Besides,

---

[1]The details of applying the superscript $z^i$ are discussed in IPNN Anonymous (2024).

---

**Algorithm 1** CIPNN or CIPAE training

---

**Input**: A sample $x_t$ from mini-batch
**Parameter**: Latent variables dimension $N$, forget number $T$, Monte Carlo number $C$, regularization factor $\gamma$, stable number $\epsilon$, learning rate $\eta$.
**Output**: $P^{\mathbf{z}}(y_l \mid x_t)$

1: Declare $\Theta$ for saving some outputs.
2: **for** $k = 1, 2, \ldots$ Until Convergence **do**
3:     Save $y_l, \mu_k^i, \sigma_k^{2,i}, i = 1, 2, \ldots, N$. into $\Theta$.
4:     **if** $len(\Theta) > T$ **then**
5:         Forget: Reserve recent T elements from $\Theta$
6:     **end if**
7:     Compute inference posterior with Eq. (21): $P^{\mathbf{z}}(y_l \mid x_t)$
8:     Compute loss with Eq. (17): $\mathcal{L}(W)$
9:     Update model parameter: $W = W - \eta \nabla \mathcal{L}(W)$
10: **end for**
11: **return** model and the inference posterior

---

$$H(\varepsilon_c) = \sum_{k=t_0}^{t_1} \left( y_l(k) \cdot \prod_{i=1}^{N} \mathcal{N}\left(\mu_t^i + \sigma_t^i \cdot \varepsilon_c; \mu_k^i, \sigma_k^{2,i}\right) \right) \tag{22}$$

$$G(\varepsilon_c) = \sum_{k=t_0}^{t_1} \left( \prod_{i=1}^{N} \mathcal{N}\left(\mu_t^i + \sigma_t^i \cdot \varepsilon_c; \mu_k^i, \sigma_k^{2,i}\right) \right) \tag{23}$$

Where $t_0 = \max(1, t_1 - T)$, $t_1$ is the number of input samples, $\varepsilon_c \sim \mathcal{N}(0, 1)$. Hyperparameter T is for forgetting use, i.e., $P^{\mathbf{z}}(y_l \mid x_t)$ are calculated from the recent T samples. The detailed algorithm implementation is shown in Algorithm (1).

## 6.2 TRAINING SETUPS

By comparing CIPNN and CIPAE, we can see that they can share the same neural network for a training task. As shown in Figure 2, the latent variables of a classification task can be visualized with CIPAE, and we can also use CIPNN to evaluate the performance of an auto-encoder task.

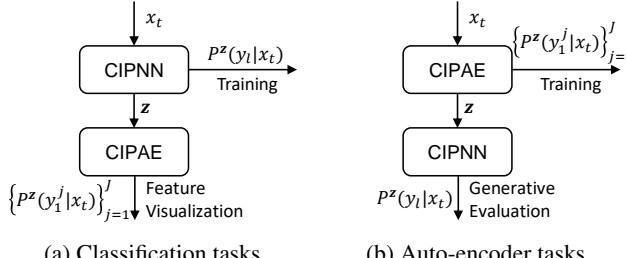

(a) Classification tasks      (b) Auto-encoder tasks

Figure 2: Training setups for classification and auto-encoder tasks. (a) CIPNN is used to do supervised classification tasks and CIPAE is used to reconstruct the input image to see what each latent variable has learned. (b) CIPAE is used to do auto-encoder task and CIPNN is used for evaluation.

## 7 EXPERIMENTS AND RESULTS

VAE validated that Monte Carlo number $C$ can be set to 1 as long as the batch size is high enough (e.g. 100) Kingma & Welling (2014), we will set batch size to 64, $C = 2$, $T = 3000$ and $\beta = 1$ for the following experiments.

## 7.1 RESULTS OF CLASSIFICATION TASKS

In this section, we use train setup in Figure 2a to perform different classification tasks in order to reconstruct the latent variable to see what they have learned.

The results from the work Anonymous (2024) show that IPNN prefers to put number 1,4,7,9 into one cluster and the rest into another cluster. We also get a similar interesting results in CIPNN, as shown in Figure 3, with stable number $\epsilon = 1$, the reconstructed image with 1-D latent space shows a strong tendency to sort the categories into a certain order and the number 1,4,7,9 stays together in the latent space. Similar results are also found with 2-D latent space, see Figure 9b. Unfortunately, we currently do not know how to evaluate this sort tendency numerically.

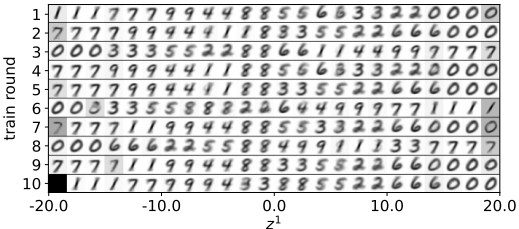

Figure 3: Reconstructed image with 1-D latent space for classification of MNIST: test accuracy is $93.3 \pm 0.5\%, \gamma = 0.95, \epsilon = 1$. The training is repeated for 10 rounds with different random seeds.

As shown in Figure 4a, with a proper regularization factor $\gamma$, the test dataset is mapped to a relative small latent space, and the over-fitting problem is avoided. Besides, in Figure 4b each joint sample area correspond to one unique category, this is consistent with our Proposition 1. In Figure 4c, the reconstructed image follows the conditional joint distribution $P\left(y_l \mid z^1, z^2\right), l = 0, 2, \ldots, 9$. Finally, we can see that the transition of one category to another is quite smooth in the latent space, this is a very good property that is worth further research in the future.

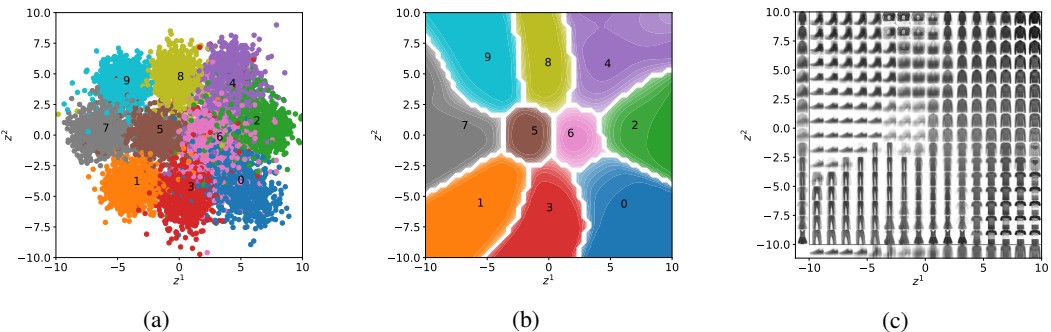

(a)                              (b)                              (c)

Figure 4: Classification Results of 2-D latent space on Fashion-MNIST: accuracy $87.6\%$, $\gamma = 0.9$, $\epsilon \approx 0$. (a) Results of latent variables on test dataset; (b) Conditional probability distribution of each category $P\left(y_l \mid z^1, z^2\right), l = 0, 2, \ldots, 9$. Colors represent probability value: from 1-dark to 0-light; (c) Reconstructed image with $(z^1, z^2)$, see Eq. (18), and image on x and y axes is reconstructed with $z^1$ and $z^2$, respectively, see Eq. (19).

**Results of Classification Tasks on Large Latent Space**   with the modification method discussed in Appendix A show that CIPNN is robust for large latent space.

**Results of Classification Tasks on more Datasets**   Further results in Table 3 on MNIST Deng (2012), Fashion-MNIST Xiao et al. (2017), CIFAR10 Krizhevsky et al. (2009) and STL10 Coates et al. (2011) show that our proposed indeterminate probability theory is valid, the backbone between CIPNN and IPNN Anonymous (2024) is the same.

## 7.2 RESULTS OF AUTO-ENCODER TASKS

Table 2: Average test accuracy of 10 times results on Large Latent Space on MNIST.

| Latent space | 5-D | 10-D | 20-D | 50-D | 100-D | 200-D | 500-D | 1000-D |
|---|---|---|---|---|---|---|---|---|
| IPNN | 94.8 | 88.6 | 80.6 | - | - | - | - | - |
| CIPNN | 95.6 | 94.7 | 94.7 | 94.9 | 94.9 | 94.9 | 94.7 | 93.4 (2 times) |

In this section, we will make a comparison between CIPAE and VAE Kingma & Welling (2014), the latter one also using train setup in Figure 2b. For VAE model, we combine it with CIPNN to evaluate its performance. Besides, the regularization loss of VAE is switched to our proposed loss, see Eq. (16). As shown in Figure 5, the results of auto-encoder tasks between CIPAE and VAE are similar, this result further verifies that CIPAE is the analytical solution.

Table 3: Test accuracy with 3-D latent space; backbone is FCN for MNIST and Fashion-MNIST, Resnet50 He et al. (2016) for CIFAR10 and STL10.

| Dataset | CIPNN | IPNN | Simple-Softmax |
|---|---|---|---|
| MNIST | $95.9 \pm 0.3$ | $95.8 \pm 0.5$ | $97.6 \pm 0.2$ |
| Fashion-MNIST | $85.4 \pm 0.3$ | $84.5 \pm 1.0$ | $87.8 \pm 0.2$ |
| CIFAR10 | $81.3 \pm 1.6$ | $83.6 \pm 0.5$ | $85.7 \pm 0.9$ |
| STL10 | $92.4 \pm 0.4$ | $91.6 \pm 4.0$ | $94.7 \pm 0.7$ |

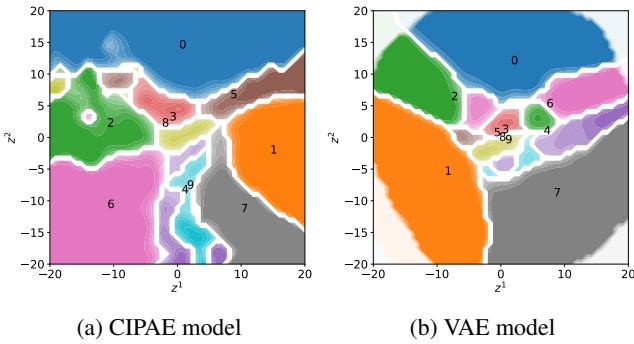

(a) CIPAE model    (b) VAE model

Figure 5: Auto-encoder results of 2-D latent space evaluated with CIPNN model on MNIST: test accuracy is $70.1\%$ for CIPAE and $67.4\%$ for VAE, $\gamma = 0.98, \epsilon \approx 0$. Conditional probability distribution of each category $P\left(y_l \mid z^1, z^2\right), l = 0, 2, \ldots, 9$. Colors represent probability value: from 1-dark to 0-light.

## 8   CONCLUSION

General neural networks, such as FCN, Resnet He et al. (2016), Transformer Vaswani et al. (2017), can be understood as a complex mapping function $f : X \rightarrow Y$ Roberts et al. (2022), but they are black-box for human Buhrmester et al. (2019). Our proposed model can be understood as two part: $f : X \rightarrow \mathbf{z}$ and $P(Y \mid \mathbf{z}) : \mathbf{z} \rightarrow Y$, the first part is still black-box for us, but the latter part is not unknown anymore. Such kind of framework may have two advantages: the first part can be used to detect the attributes of datasets and summarize the common part of different categories, as shown in Figure 3; the latter part is a probabilistic model, which may be used to build a large Bayesian network for complex reasoning tasks.

Besides, our proposed framework is quite flexible, e.g. from $X$ to $\mathbf{z}$, we can use multiple neural networks with different structures to extract specific attributes as different random variables $z^i$, and these random variables will be combined in the statistical phase.

Although our proposed model is derived from indeterminate probability theory, we can see Determinate from the expectation form in Eq. (11). Finally, we'd like to finish our paper with one sentence:

The world is determined with all Indeterminate!

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

## A  LARGE LATENT SPACE

Although Eq. (2) has no limitation on the number of latent variables, CIPNN cannot use two many variables due to the limitation of Pytorch float32 range, this will cause the term from Eq. (22) and Eq. (23) to

$$float32\left(\prod_{i=1}^{N}\mathcal{N}\left(\mu_t^i + \sigma_t^i \cdot \varepsilon_c; \mu_k^i, \sigma_k^{2,i}\right)\right) = 0 \text{ or } \inf \text{ , for large N,} \tag{24}$$

where $\varepsilon_c \sim \mathcal{N}(0,1)$.

Because of this issue, CIPNN/CIPAE can only support a maximum of 30 to 50 latent variables. To solve this problem, we introduce a parameter $\alpha$ to Eq. (22) and Eq. (23) as

$$H(\varepsilon_c) = \sum_{k=t_0}^{t_1}\left(y_l(k) \cdot \prod_{i=1}^{N}\left(\alpha^i \cdot \mathcal{N}\left(\mu_t^i + \sigma_t^i \cdot \varepsilon_c; \mu_k^i, \sigma_k^{2,i}\right)\right)\right) \tag{25}$$

$$G(\varepsilon_c) = \sum_{k=t_0}^{t_1}\left(\prod_{i=1}^{N}\left(\alpha^i \cdot \mathcal{N}\left(\mu_t^i + \sigma_t^i \cdot \varepsilon_c; \mu_k^i, \sigma_k^{2,i}\right)\right)\right) \tag{26}$$

Where $\alpha^i > 0$ is a scaler constant value, and it can be easily proved that this value will not affect the result of Eq. (21).

How to get a proper scaler value $\alpha^i$? Let

$$v_k^i = \frac{\mu_t^i + \sigma_t^i \cdot \varepsilon_c - \mu_k^i}{\sigma_k^i} \tag{27}$$

$$\Rightarrow \varepsilon_c = \frac{v_k^i - \frac{\mu_t^i - \mu_k^i}{\sigma_k^i}}{\sigma_t^i / \sigma_k^i} \sim \mathcal{N}(0,1), \tag{28}$$

so $v_k^i \sim \mathcal{N}\left(\frac{\mu_t^i - \mu_k^i}{\sigma_k^i}, \sigma_t^i / \sigma_k^i\right)$. Then we get

$$\frac{1}{N}\sum_i^N v_k^{2,i} \approx \mathbb{E}\left[v_k^{2,i}\right] = \left(\frac{\mu_t^i - \mu_k^i}{\sigma_k^i}\right)^2 + \left(\sigma_t^i / \sigma_k^i\right)^2 = 1, \text{ for } \mu_k^i = \mu_t^i \text{ and } \sigma_k^i = \sigma_t^i. \tag{29}$$

Where $\mu_k^i = \mu_t^i$ and $\sigma_k^i = \sigma_t^i$ is the critical case. The product term from Eq. (25) and Eq. (26) can be written as

$$\prod_{i=1}^{N}\left(\alpha^i \cdot \mathcal{N}\left(\mu_t^i + \sigma_t^i \cdot \varepsilon_c; \mu_k^i, \sigma_k^{2,i}\right)\right) = \prod_{i=1}^{N}\left(\alpha^i \cdot \frac{1}{\sigma_k^i \sqrt{2\pi}} \cdot e^{-\frac{1}{2}v_k^{2,i}}\right) \tag{30}$$

$$= \prod_{i=1}^{N}\left(\alpha^i \cdot \frac{1}{\sigma_k^i \sqrt{2\pi}}\right) \cdot e^{-\frac{1}{2}\sum_i^N v_k^{2,i}} \tag{31}$$

$$\approx \prod_{i=1}^{N}\left(\alpha^i \cdot \frac{1}{\sigma_k^i \sqrt{2\pi}}\right) \cdot e^{-\frac{1}{2}N \cdot \mathbb{E}\left[v_k^{2,i}\right]} \tag{32}$$

$$= \prod_{i=1}^{N}\left(\alpha^i \cdot \frac{1}{\sigma_k^i \sqrt{2\pi}}\right) \cdot e^{-\frac{1}{2}N}, \text{ for } \mu_k^i = \mu_t^i, \sigma_k^i = \sigma_t^i. \tag{33}$$

Let Eq. (33) be 1, so we have

$$\prod_{i=1}^{N}\left(\alpha^i \cdot \frac{1}{\sigma_k^i \sqrt{2\pi}} \cdot e^{-\frac{1}{2}}\right) = 1 \tag{34}$$

$$\Rightarrow \alpha^i \cdot \frac{1}{\sigma_k^i \sqrt{2\pi}} \cdot e^{-\frac{1}{2}} = 1 \tag{35}$$

$$\Rightarrow \alpha^i = \sigma_k^i \sqrt{2\pi e} \tag{36}$$

$$\Rightarrow \alpha^i = \sqrt{2\pi e}, \text{ for } \sigma_k^i = 1. \tag{37}$$

Because $\alpha^i$ must be invariant over different $k = 1, 2, \ldots, n$, otherwise, the results of Eq. (21) will be affected. For CIPNN/CIPAE the variance is around 1 due to the regularization loss term, see Eq. (16), so we set $\alpha^i = \sqrt{2\pi e} \approx 4.1327313541$.

In this way, we can now apply CIPNN/CIPAE to a maximum of thousands of latent variables, see Table 2. For a larger latent space, we will encounter this issue again, and it needs to be solved in the future.

## B  VISUALIZATION

With the visualization method proposed in Eq. (19), we can see what each latent variable has learned in the 10-D latent space. As shown in Figure 6, each latent variable focuses on mainly one or two different categories.

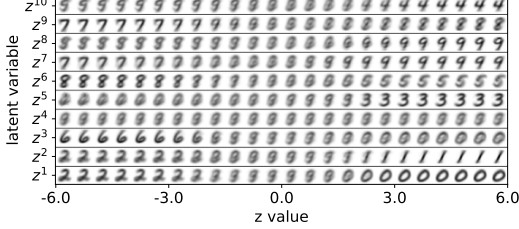

Figure 6: Classification results of 10-D latent space: test accuracy is $97.1\%, \gamma = 0.8, \epsilon \approx 0$. Images are reconstructed with one latent variable $z^i, i = 1, 2, \ldots, 10$, see Eq. (19).

Figure 7 shows classification results of 20-D latent space on Dogs-vs.-Cats-Redux dataset, we can see that each latent variable focuses on both two different categories.

Impact analysis of regularization factor $\gamma$ is discussed in Figure 8.

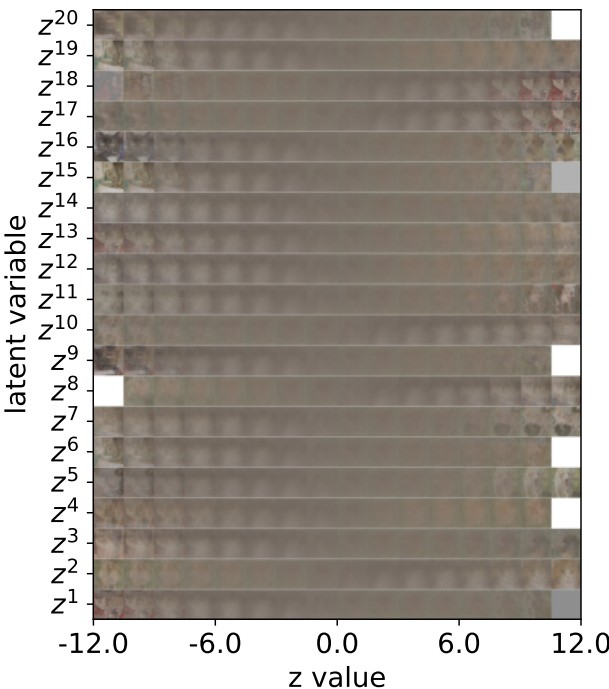

Figure 7: Classification results of 20-D latent space on Dogs-vs.-Cats-Redux: test accuracy is $95.2\%, \gamma = 0.999, \epsilon \approx 0$. Images are reconstructed with one latent variable $z^i, i = 1, 2, \ldots, 20$, see Eq. (19).

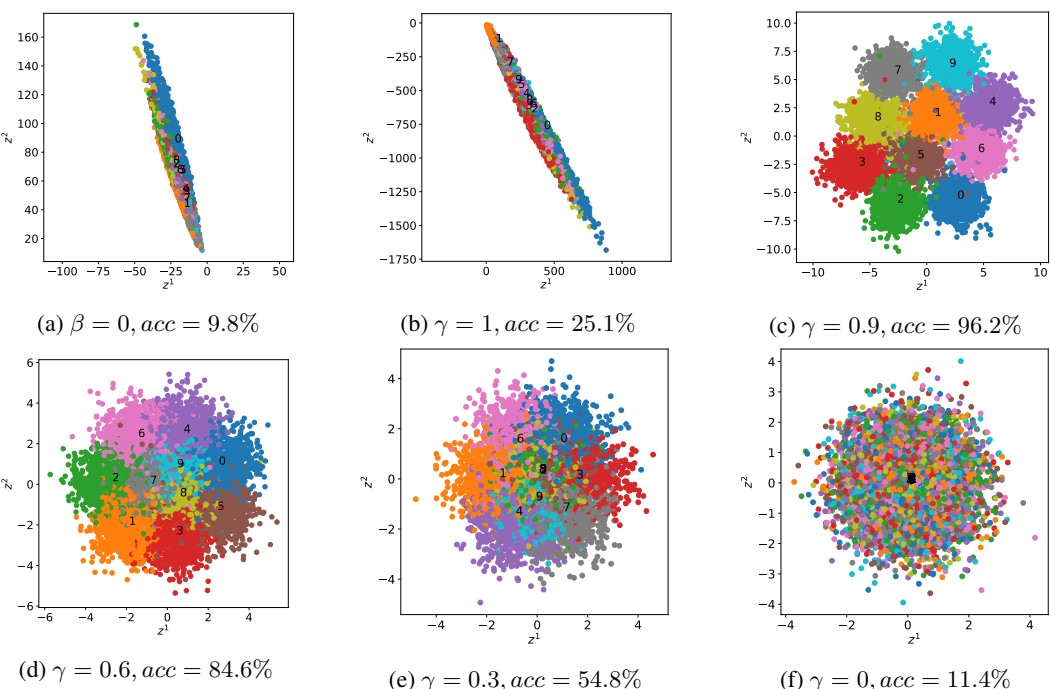

(a) $\beta = 0, acc = 9.8\%$     (b) $\gamma = 1, acc = 25.1\%$     (c) $\gamma = 0.9, acc = 96.2\%$

(d) $\gamma = 0.6, acc = 84.6\%$     (e) $\gamma = 0.3, acc = 54.8\%$     (f) $\gamma = 0, acc = 11.4\%$

Figure 8: Impact analysis of regularization factor $\gamma$ with 2-D latent space of classification results on MNIST. (a) Without regularization term, the model has a strong over-fitting problem (train accuracy is around $100\%$), and the variance is near to 0, that's why it's space is not big than (b). (b) The variance is around 1, however, without constrain the distribution to be connected together, the space is very high, and it also shows over-fitting problem. (c) With proper $\gamma$ value, we get an optimal effective latent space. (d,e,f) As $\gamma$ value further reduces, the latent space is getting smaller, and both tain and test accuracy is reducing due to over regularization.

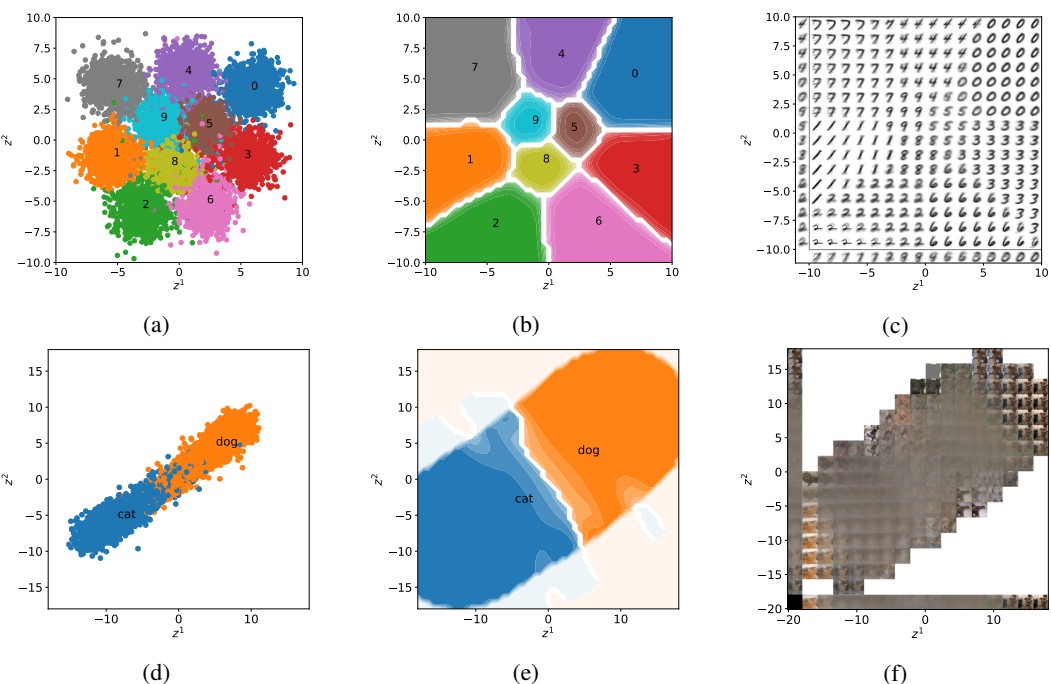

Figure 9: Classification Results of 2-D latent space on MNIST and Dogs-vs.-Cats-Redux: test accuracy is $96.1\%$ and $95\%$, respectively. $\gamma = 0.9$ for MNIST, $\gamma = 0.9999$ for Dogs-vs.-Cats-Redux, $\epsilon \approx 0$. (a,d) Results of latent variables on test dataset; (b,e) Conditional probability distribution of each category $P\left(y_l \mid z^1, z^2\right), l = 0, 2, \ldots, 9$. Colors represent probability value: from 1-dark to 0-light; (c,f) Reconstructed image with $(z^1, z^2)$, see Eq. (18), and image on x and y axes is reconstructed with $z^1$ and $z^2$, respectively, see Eq. (19).

