# OpenReview forum: "Continuous Indeterminate Probability Neural Network"
_ICLR.cc/2024/Conference — Submitted to ICLR 2024_

### Official Review · Reviewer_7gbu · 2023-10-21

**Soundness:** 2 fair
**Presentation:** 1 poor
**Contribution:** 2 fair
**Rating:** 3
**Confidence:** 2

**Summary:**

The paper introduces Continuous Indeterminate Probability Neural networks, which applies Indeterminate Probability Theory to define neural networks with latent variables for classification.
The authors also present a related auto-encoding variant of the model, which can be used to visualize the latent variable of the model.

**Strengths:**

* The ideas presented in the paper are interesting and novel, to the best of my knowledge. These ideas could inspire future research.

* Due to the usage of the latent variables in the CIPNN, the proposed model is less black-box than other architectures

**Weaknesses:**

**General comments**

There are 2 major issues with this paper, regarding clarity and experiments.

CLARITY

Overall, I found the paper hard to understand, mostly because it relies heavily on the unpublished work in (Anonymous, 2024), and assumes that the reader is knowledgeable of its content.
While there is a high level description of (Anonymous, 2024) in section 2.2, this description is rushed and confusing (see detailed comments below).

Being a conference paper, this paper should instead be self-contained: the reader/reviewer should not have to read in full (Anonymous, 2024) to understand the proposed method (especially keeping in mind that the other paper could be rejected from the conference and be therefore unpublished if this work gets accepted).
As is, this paper looks more like an appendix to (Anonymous, 2024), rather a paper by itself. I suggest that the authors read and rewrite this work with the eyes of someone that knows nothing about (Anonymous, 2024).

Considering the classification/auto-encoding applications of Indeterminate Probability Theory, there are also several points in the paper that need to be clarified/improved.

EXPERIMENTS

The experimental section is also quite confusing, and lacks proper baselines to understand the real performances of the model.



**Detailed comments**

Below I describe the main points of confusion in each section.

_Abstract_

You write "pushed this classification capability to infinity" -> what does this mean?

_Introduction_

The motivation for this work in the introduction is based on the IPNN, which is however a model the reader knows nothing about at this point in the paper.

_Section 2.1_

You write "VAE uses neural network as the approximate solution of decoder"
 -> What does this mean? In a VAE the decoder is defined as a modelling choice, and the encoder is used to approximate the posterior probability.

_Section 2.2_

Overall this section is hard to understand, and needs a better example/toy problem to help the reader (you could focus on the classification task from Figure 1 for example).

When you say "introducing Observers and treating the outcome of each random experiment as indeterminate probability distribution,"
- What are Observers? They are no longer mentioned in the rest of the section
- how do you define an "indeterminate" probability distribution?

These definitions are missing:
- what does $m$ represent in $y_m$
- what does $l$ represent in $y_l$
- what does $t$ represent in $x_t$


_Section 3_

Why do you need to introduce both Observer 2 and Observer 3? What is the difference? Observer 2 seems not to be relevant for the subsequent discussion.
Due to the confusion in Section 2.2, I am not really sure what you are trying to achieve in this section, and how exactly this relates to the rest of the paper.



_Section 4.2_

- Why did you choose that specific distribution in the right-hand side of the KL divergence?

_Section 5_

You refer to details in (Anonymous, 2024) in the footnote, but they are needed in this paper as well to understand it.


_Section 6_

"In this section, we will focus on the training strategy of Gaussian distribution" ->
Can you clarify what this means?


_Section 7.1_

1. This section misses baselines for other classification models (even simple neural networks)? The classification performances of your model on MNIST look quite poor for example.
1. In Table 3 you compare against "Simple-Softmax", which is not defined, and which performs significantly better than the proposed model
1. The advantages of this model vs other architectures are not well described
1. What's the scalability of this method? What are the training times?
1. The dataset names are not even mentioned in the main text, so one needs to guess which dataset the authors are talking about while reading this section. Only captions in the Figures mention the dataset.
1. In the paragraph "Results of classification tasks on large latent spaces" - are you talking about table 2? It is not mentioned


_Section 7.2_

1. The difference between CIPAE and VAE is not clear from the paper
1. "As shown in Figure 5, the results of
auto-encoder tasks between CIPAE
and VAE are similar, this result further verifies that CIPAE is the analytical solution." -> What does this mean? Why can you make this statement from looking at a Figure?

_Conclusion_

"Although our proposed model is derived from indeterminate probability theory, we can see Determinate
from the expectation form in Eq. (11). Finally, we’d like to finish our paper with one sentence:
The world is determined with all Indeterminate!" -> not sure what this means.

**Questions:**

See the questions in the above section.

---

> ### Author Response · Authors · 2023-11-21
> **Response to Reviewer 7gbu**
>
> Dear Reviewer 7gbu,
>
> We sincerely appreciate your very insightful and detailed feedbacks. Here are our detailed responses to your questions.
>
> **Q1**: You write "pushed this classification capability to infinity" -> what does this mean?
>
> A1: It means that the model is very flexible to classification tasks, we can use the same neural network without changing the number of output nodes to classify arbitrary categories. For example, in Fig. 9, we used two random variables (4 nodes in the last layer) to classify into 2 or 10 categories. And we can also use same model to classify a very large number of categories without limitation.
>
> **Q2**: The motivation for this work in the introduction is based on the IPNN, which is however a model the reader knows nothing about at this point in the paper.
>
> A2: We apologize for the additional reference to our other paper, which may have disrupted your reading.
>
> **Q3**: You write "VAE uses neural network as the approximate solution of decoder" -> What does this mean?
>
> A3: It means that the decoder of VAE is a model, but the decoder of our CIPAE is a probability equation, that means we do not need to design the decoder.
>
> **Q4**: Overall Section 2.2 is hard to understand, and needs a better example/toy problem to help the reader.
>
> A4: Actually, the toy example in Section 3 is a very good toy example to CIPNN.
>
>
> **Q5**: What are Observers? Why do you need to introduce both Observer 2 and Observer 3? What is the difference?.
>
> A5: In CIPNN, the label can be understood as Observer 1, the model output can be understood as Observer 3.
> (Observer 2 is the case for IPNN.)
>
> In addition, we'd like to answer this question from a high-level.
>
> The world is understood through observers. In the case of a coin toss, the outcome cannot be known unless observed. That is, the ground truth is not able to be known, we only know the observations.
>
> Given that the use of observers is unavoidable, and the imperfect observations maybe more general in the real world? (Perfect observations are the special case.)
>
> Additionally, we do not need to restrict the observers, as they may have different ways of understanding the world. For example, in our coin toss example in Section 3, Observer$_3$ understands the outcome as a Gaussian distribution.
>
> The questions now is: what truly matters?
>
> The Change! While different observers may have different understandings of the world, the Change still follows the Ground Truth (albeit with some errors). Our proposed equation is designed to evaluate this Change.
>
> **Q6**: Why did you choose that specific distribution in the right-hand side of the KL divergence?
>
> A6: As we written in the paper, KL divergence is for solving over-fitting problem, and our modification is according to the analysis results in Fig. 8.
>
> **Q7**: "In this section, we will focus on the training strategy of Gaussian distribution" -> Can you clarify what this means?
>
> A7: It means that using a distribution other than Gaussian, such as a uniform distribution or others, is also acceptable.
>
> **Q8**: This section misses baselines for other classification models (even simple neural networks)?The classification performances of your model on MNIST look quite poor for example.
> In Table 3 you compare against "Simple-Softmax", which is not defined, and which performs significantly better than the proposed model
>
> A8: Our proposed model is not for a performance improving, as you can see its performance is even poor than the most simple neural networks, so it doesn't need to compare it with other better models.
>
> **Q9**: "As shown in Figure 5, the results of auto-encoder tasks between CIPAE and VAE are similar, this result further verifies that CIPAE is the analytical solution." -> What does this mean? Why can you make this statement from looking at a Figure?
>
> A9: CIPAE is the analytical solution is a theoretical analysis results, and the results in Fig. 5 is an evidence.
>
>
> **Q10**: "Although our proposed model is derived from indeterminate probability theory, we can see Determinate from the expectation form in Eq. (11). Finally, we’d like to finish our paper with one sentence: The world is determined with all Indeterminate!" -> not sure what this means.
>
> A10: This is an abstract understanding of Eq. (11).
>
> Best regards
>
> Authors

---

> > ### Comment · Reviewer_7gbu · 2023-11-22
> > **Unchanged score**
> >
> > Thanks for the response to my comments.
> >
> > While some of the points were clarified in my response, overall I believe that the required changes to the paper in terms of clarity will need a new review cycle. Hopefully by then (Anonymous, 2024) will be published, and it will be much easier for you to describe the motivation for this work (as also noted by reviewer EMUS) and the CIPNN model.
> >
> > For now I will then leave the score unchanged, but will be open to increase it in case other reviewers convince me during the reviewer discussion phase.

---

### Official Review · Reviewer_Vwsb · 2023-10-30

**Soundness:** 3 good
**Presentation:** 2 fair
**Contribution:** 3 good
**Rating:** 8
**Confidence:** 3

**Summary:**

The paper “Continuous indeterminate…” proposes a continuous extension of the “Indeterminate …” model by the same authors, correctly referenced as Anonymous. The paper describes this extension, accompanied by definitions of the classification and auto-encoder models, together with training, inference procedures and simple experiments. The resulting models are only a bit less accurate than well known models. The author's main goal is to theoretically describe and show the benefits from its use.

In my opinion, the paper may be accepted, provided the authors answer the above doubts.

**Strengths:**

1. The model shown is interesting and shows, perhaps not very illuminating but still explainable to the input —> latent —> classification, and input —> latent —> reconstruction problems in theoretical way.
2. There is a good introductory to the theory in section 3.
3. The proposed model aims at providing more explainable solutions to classification, although there is some way before the model may accomplish that.
4. The performed experiments prove, or at least show, the hypothesis clearly stated by the authors.

**Weaknesses:**

1. I guess the whole paper should start with a deeper explanation of differences between the proposed approach and a VAE model.
2. An ablation study concerning the complexity is missing. The authors say that the C hyperparameter can be set to 1 “as long as the batch size is high enough…” (page 7, bottom), but still they use C=2 in the experiments.
3. The derivation for the continuous probability mathematical formulae is complex, and lacks intuition, instead giving intricate formulas and variables.
4. Although the authors correctly reference to their own paper as written by Anonymous, but not yet published. The authors do it all through the paper referencing the reader to find details over there. The paper can easily be found by the title. On the other hand, this is unavoidable.

**Questions:**

1.  Add some introduction to differences between the proposed model and VAE-like approaches, or perhaps accompany the whole sequence should be accompanied by comparisons to corresponding steps in a VAE-type model?
2. Equation (21), being the basis for training, needs a deeper explanation. Why use the max functions both in numerator and denominator?
3. When comparing the proposed CIPAE with VAE, section 7.2, the visualizations of the latent space become somehow different, with parts of the R^2 latent for VAE empty. Does it come from different latent definition in both cases? Or is it just a result of showing only the [-20, 20]x[-20,20] square? This might not seem to be a fair comparison unless explained.
4. From a practical point of view: what is the training and inference comparison between VAE (as well as other WAE, etc.) approach and the proposed ones?
5.  What is the impact of C value and the batch size on quality, trading and inference time, etc.? Could you provide some ablation study?
6. In conclusion the authors state, that the proposed model is actually composed of two parts: first detects attributes, and the second (i.e. classification?) is a probabilistic model which may be used for reasoning. A 1-D example shown in figure 3, that the authors refer to, shows that the first part performs a kind of clustering, is that so? Please elaborate on that, since it would greatly enlarge the readability of the paper.
7. Several language errors should be corrected. E.g. a) on page 1 the sentence “However, IPNN need to predefine the …” should probably be “However, IPNN needs to be predefined…”; b) just above Eq. 1 instead “bellow” should be “below”; c) what is the word “complexer” at the bottom of page 3? Perhaps the authors meant to say “more complex? “Complexer” might be used in French. I would suggest checking the whole text with some native speaker. These errors are usually tiny, but disturb reading.
8. Please, if possible, make the figures a bit larger, just to make them somehow more readable. This refers particularly to figures 1, 3, and perhaps 2 and 4 too.
9. In section 7.1 you claim that the CIPNN tends to put 1, 4, 7, 9 MNIST numbers in one cluster — this is hardly visible in the figures. How is that model used for classification? Which inputs were used in each round? Could you elaborate on that a bit?
10. Equations are sometimes complex (lots of variables and indices), e.g. Sequence from (9) to (14). Could you, please, make them easier to follow?
11. Some small editing errors, e.g. subsection 7.2 title starts as an orphan.

---

> ### Author Response · Authors · 2023-11-18
> **Response to Reviewer Vwsb (1/2)**
>
> Dear Reviewer Vwsb,
>
> We sincerely appreciate your very insightful and detailed feedbacks, and we also thank you very much for your understanding of the unavoidable reference to our other paper. Here are our detailed responses to your questions.
>
> **Q1**: Add some introduction to differences between the proposed model and VAE-like approaches
>
> A1: OK. The framework of CIPAE and VAE are very similar, but the two models are derived from different theories.
> 1. The KL regularization term is an irremovable part in ELBO. Our KL regularization term is additional for over-fitting problem, and it is allowed to use other regularization method for CIPAE.
> 2. If we use a model as the decoder for CIPAE (not Eq. (1) in the paper), here is the detailed difference:
>
> **VAE**:
>
> $$ ELBO =-D_{KL}(q_{\phi}(\mathbf{z}\mid x^{(i)}) \parallel p_{\theta}(\mathbf{z})) + E_{q_{\phi}(\mathbf{z}\mid x^{(i)})} \left[\log p_{\theta}(x^{(i)}\mid \mathbf{z}) \right] \text{   (a)}$$
>
>
> **Approximate CIPAE** (Not the original CIPAE):
> $$
> \underset{\text{(b}}{\underbrace{\log P^{\mathbf{z}}(y \mid x_{t})}}
> =\underset{\text{(c)}}{\underbrace{\log \int\limits_{\mathbf{z}}P(y,\mathbf{z}\mid x_{t})}}
> =\underset{\text{(d)}}{\underbrace{\log \int\limits_{\mathbf{z}}\left ( P_{\theta}(y\mid\mathbf{z})
> \cdot P_{\phi}(\mathbf{z}\mid x_{t}) \right )}}
> = \underset{\text{(e)}}{\underbrace{\log E_{P_{\phi}(\mathbf{z}\mid x_{t})}
> \left[P_{\theta}\left(y\mid \mathbf{z}\right)\right] }}
> $$
> From (b) to (c): marginalization.
> From (c) to (d): according to axiom 3 that given $\mathbf{z}$, $x$ and $y$ (image pixels) is mutual independent.
> From (d) to (e): the decoder is approximated with a model $\theta$.
>
> If we set the regularization term of CIPAE to be the same as that of VAE, our loss will be almost identical to that of VAE. The only difference is that the $\log$ term is outside of the expectation term $\mathbb{E}$. For the case where the Monte Carlo number C = 1, the losses of VAE and approximate CIPAE will be exactly the same.
>
> This analysis results are quite interesting, that VAE and approximate CIPAE are derived from very different theory, however their final code implementations can be same in a special case (C=1).
>
> **Q2**: Equation (21), being the basis for training, needs a deeper explanation. Why use the max functions both in numerator and denominator?
>
> A2: Actually, the max functions are not important, because we usually set $\epsilon=10^{-20}$. Therefore, the max functions are not activated for the importantsample space.
>
> **Q3**: ...the visualizations of the latent space become somehow different, with parts of the R^2 latent for VAE empty.
>
> A3: You can get a sense of the difference by comparing Fig. 4a and Fig. 4b. In Fig. 4a, the effective points are confined to a small space, whereas the valid space of CIPNN is larger, as shown in Fig. 4b.
> Similarly, for CIPAE, although it is valid in a larger space, the values of Eq. (22) and Eq. (23) are very small, and the test or train points will not fall within that region.
>
> **Q4**: From a practical point of view: what is the training and inference comparison between VAE and CIPAE?
>
> A4: We are afraid that we cannot finished it during rebuttal. Maybe after rebuttal?

---

> > ### Author Response · Authors · 2023-11-18
> > **Response to Reviewer Vwsb (2/2)**
> >
> > **Q5**: What is the impact of C value and the batch size on quality, trading and inference time, etc.? Could you provide some ablation study?
> >
> > A5: Thank you for you suggestions, here are our ablation study results on MNIST with 100-D latent space.
> >
> > |Monte Carlo\Batch size|1|8|32|64|
> > |-|-|-|-|-|
> > |1|85.6|90.1|91.9|92.9|
> > |2|90.3|94.5|95.5|95.7|
> > |8|94.9|96.6|97.4|97.5|
> > |16|95.8|97.2|97.5|97.5|
> >
> > where regularization factor $\gamma = 0.7$ for the batch size = 1, $\gamma =0.8$ for other tests, forget number $T=500$. And mean value of 3 times tests is reported (Fewer points test only done 1 times due to too slowly.).
> >
> > As we can see from the above table, we can set a small Monte Carlo number for big batch size.
> >
> > **Q6**: In conclusion ...first part is for detects attributes... A 1-D example shown in figure 3, shows that the first part performs a kind of clustering, is that so?
> >
> > A6: Suppose we need to classify a large number of images across an extremely large number of categories, and the latent space is multi-dimensional. It becomes important if one random variable can summarize the attributes of the image, for example, the model learned that $z^{1}$ is for color, $z^{2}$ is for shape, and so on. In this way, the large joint space will be more efficient, because each $z$ is for special information. And in our opinion, clustering during classification is kind of summarizing the image attributes.
> >
> >
> > **Q7**: Several language errors should be corrected.
> >
> > A7: Thank you very much for your help, it is very nice of you for such kind of suggestions. We will correct them according to your suggestions.
> >
> > **Q8**: Please, if possible, make the figures a bit larger...
> >
> > A8: We also want larger figures, but it will make our contents more than 9 pages, this is a regrettable compromise.
> >
> > **Q9**: In section 7.1 you claim that the CIPNN tends to put 1, 4, 7, 9 MNIST numbers in one cluster — this is hardly visible in the figures. How is that model used for classification?
> >
> > A9: The model and settings (with different random seed) are the same for each round of classification.
> >
> > In addition, we reorganized the results of Fig. 3 with the following table by ignoring the value direction of $z^{1}$.
> >
> > |Round| Cluster1|Cluster2|
> > |-|-|-|
> > |1|1,7,9,4|8,5,6,3,2,0|
> > |2|7,9,4,1|8,3,5,3,6,0|
> > |3|7,9,4,1|6,8,2,5,3,0|
> > |4|7,9,4,1|8,5,6,3,2,0|
> > |5|7,9,4,1|8,3,5,2,6,0|
> > |6|1,7,9,4|6,2,8,5,3,0|
> > |7|1,7,9,4|8,5,6,3,2,0|
> > |8|7,3,1,9,4|8,5,2,6,0|
> > |9|7,1,9,4|8,3,5,2,6,0|
> > |10|1,7,9,4|3,8,5,2,6,0|
> >
> > As you can see, '1,7,9,4' is clustered $9/10$ times into one cluster.
> >
> > **Q10**: Equations are sometimes complex (lots of variables and indices), e.g. Sequence from (9) to (14). Could you, please, make them easier to follow?
> >
> > A10: Thank you for your reminding. However, as you can see, our proposed indeterminate equation in IPNN is also complex to read.
> >
> > **Q11**: Some small editing errors, e.g. subsection 7.2 title starts as an orphan.
> >
> > A11: Thank you very much again.
> >
> >
> > Best regards
> >
> > Authors

---

> > > ### Comment · Reviewer_Vwsb · 2023-12-02
> > > **No assessment upgrade**
> > >
> > > Thank you for the answers
> > >
> > > Q-A 1: shouldn't your answer above be added to the text, apart from your last paragraph of subsection 2.1 (VAE) to be clear for all the readers? As for the space needed, I believe that the example in section 3, being simplistic, could be much shortened.
> > >
> > > Q-A 2, 3, 4: OK, I can understand.
> > >
> > > Q-A 5: thank you for the assessment, but why don't you add that conclusion, given in the answer above, to the text?
> > >
> > > Q-A 6: I like the inference, or rather the objective, presented in the first paragraph of the conclusion. On the other hand, I cannot agree with the reasoning as presented above in the answer that the found variables z_1, …, z_N represent simply interpretable features as presented in the answer. It can be postulated that the found variables z_k are orthogonal, but there is still no evidence that this is indeed the case in your model. This is still the essence of my question as it is not apparent and the reasoning is a bit of a wishful thinking type.
> > > Still, do you postulate that your model produces a latent space of z_k variables, or even more, that z_k explains more variance (or other measure) than z_{k+1} and so on?
> > >
> > > Q-A 7: OK
> > >
> > > Q-A 8: I still believe some images could be made larger, e.g. by joining some -- e.g. figs. 3 & 5 and using up the saved space. Still, the objective is not to fit onto 9 pages, but present your model clearly ;-) On the other hand, viewing some images in colour helps.
> > >
> > > Q-A 9: Have you uploaded a new version of your paper, or is the fig. 3 unchanged? In the version to download, I can see no difference. The way to read this figure is still not very clear to me.
> > >
> > > Q-A 10, 11: OK.
> > >
> > > Perhaps you should note that your proposition is more on the theoretical side? There are still several points that are on rather the wishful side, and I have probably been too optimistic in my first assessment.
> > >
> > > Thus, I cannot upgrade my assessments.

---

### Official Review · Reviewer_EMUS · 2023-10-31

**Soundness:** 1 poor
**Presentation:** 1 poor
**Contribution:** 1 poor
**Rating:** 3
**Confidence:** 3

**Summary:**

This paper proposes a general model called CIPNN - Continuous Indeterminate Probability Neural Network using a group of reference variables $z$.

**Strengths:**

This paper focuses on the explainability of probabilistic models and neural networks, which is an interesting and important topic.

**Weaknesses:**

1. I find the motivation of this paper unclear. I had difficulty following the progression from Section 2.2 to Section 3 and then to the CIPNN model.

2. The indeterminate probability theory is not surprising to me, and I believe it can be easily derived via the definition of conditional probability. Equations (1) and (2) hold for any $z$, but it's not clear to me which specific type of $z$ we are expecting in the learning process.

3. I find Proposition 1 to be weak from my perspective. Specifically, Proposition 1 states, ''If $P(y_l | z^1, ... , z^N) \to \infty$, CIPNN converges to the global minimum.'' This is equivalent to saying that successful classification depends on our ability to learn a set of favorable variables, namely $z^1, ... , z^N$. However, the main challenge lies in determining the existence of these 'good' variables and how we can identify and obtain such $z^1, ... , z^N$ with theoretical guarantees.

4. I did not find the comparison with existing approaches. Also, the numerical results did not indicates the improved performance is indeed from the introduction of $z^1, ... , z^N$.

**Questions:**

see weakness.

---

> ### Author Response · Authors · 2023-11-21
> **Response to Reviewer EMUS**
>
> Dear Reviewer EMUS,
>
> We sincerely appreciate your insightful feedbacks. Here are our detailed responses to your questions.
>
> **Q1**: I find the motivation of this paper unclear. I had difficulty following the progression from Section 2.2 to Section 3 and then to the CIPNN model.
>
> A1: We apologize for the additional reference to our other paper, which may have disrupted your reading.
>
> **Q2**: The indeterminate probability theory is not surprising to me, and I believe it can be easily derived via the definition of conditional probability. Equations (1) and (2) hold for any
> , but it's not clear to me which specific type of z we are expecting in the learning process.
>
> A2: At the very least, CIPAE has a very special property:: The famous VAE model requires a decoder model, whereas the decoder of CIPAE is a probability equation, and therefore does not require a model. Have you ever seen an image auto-encoder whose decoder does not require a model?
>
> **Q3**: I find Proposition 1 to be weak from my perspective. Specifically, Proposition 1 states, ''If $P(y_l | z^1, ... , z^N) \to 1$, CIPNN converges to the global minimum.'' This is equivalent to saying that successful classification depends on our ability to learn a set of favorable variables, namely $z^1, ... , z^N$. However, the main challenge lies in determining the existence of these 'good' variables and how we can identify and obtain such $z^1, ... , z^N$ with theoretical guarantees.
>
> A3: Your understanding is correct, the classification depends on our ability to learn a set of favorable variables, namely $z^1, ... , z^N$. Firstly, through experimentation, we can see that our models are able to learn these 'good' variables. Additionally, the main loss in Eq. (15) is an application of maximum likelihood which serves as a theoretical guarantee.
>
>
> **Q4**: I did not find the comparison with existing approaches. Also, the numerical results did not indicates the improved performance is indeed from the introduction of $z^1, ... , z^N$.
>
> A4: Our proposed model is not for a performance improving, as you can see its performance is even poor than the most simple neural networks, so it doesn't need to compare it with other better models.
>
>
> Best regards
>
> Authors

---

### Meta-Review · Area_Chair_FFj7 · 2023-12-05

**Metareview:**

This paper introduces Continuous Indeterminate Probability Neural Networks, based on Indeterminate Probability Theory. While some aspects of the method seem promising, with the potential to, for example, aid interpretability, overall the paper lacks sufficient clarity to merit publication at this time. The reviewers found many aspects of the presentation confusing and difficult to digest, including both the prose and the technical equations. The heavy reliance on results from unpublished work compounded this problem. Additionally, ther reviewers noted some additional weaknesses, such as unconvincing experiments.

**Justification For Why Not Higher Score:**

The presentation lacks sufficient clarity and the experimental results are not convincing.

**Justification For Why Not Lower Score:**

N/A

---

### Decision · Program_Chairs · 2024-01-16

Reject